# Effects of Five Different Withering Methods on the Composition and Quality of Congou Black Tea

**DOI:** 10.3390/foods13213456

**Published:** 2024-10-29

**Authors:** Yamin Wu, Xinghua Wang, Lijiao Chen, Qiang Li, Junjie He, Xiujuan Deng, Jiayi Xu, Raoqiong Che, Jianyun Zhou, Wenxia Yuan, Tianyu Wu, Juan Tian, Yaping Chen, Baijuan Wang

**Affiliations:** 1College of Tea Science, Yunnan Agricultural University, Kunming 650201, China; m1678447405@163.com (Y.W.); wangaoyu2011@163.com (X.W.); 2015056@ynau.edu.cn (L.C.); 18087827443@163.com (J.H.); 15808869561@163.com (X.D.); 18088111511@163.com (J.X.); 15288131741@163.com (R.C.); sunnyaz@163.com (J.Z.); yuanwenxia2023@163.com (W.Y.); wty15508851042@163.com (T.W.); 13887152607@163.com (J.T.); 2School of Food and Biological Engineering, Chengdu University, Chengdu 610106, China; liqiang02@cdu.edu.cn; 3College of Plant Protection, Yunnan Agricultural University, Kunming 650201, China

**Keywords:** Congou black tea, withering methods, quality, nonvolatile substance, components, aroma

## Abstract

To explore the effects of different withering methods on the quality of Congou black tea, this study focused on five different withering methods: natural withering, warm-air withering, sun–natural combined withering, sun withering, and shaking withering. Gas chromatography‒mass spectrometry (GC‒MS), high-performance liquid chromatography (HPLC), and ion-exchange chromatography techniques were used to analyze the nonvolatile and volatile components and composition of the tea. The results revealed significant differences (*p* < 0.05) in the contents of nonvolatile constituents including caffeine, polyphenols, soluble sugars, free amino acids and their components, theaflavins, thearubigins, and catechins among the five different withering methods, with varying degrees of correlation between these components. A total of 227 aroma compounds were detected, and significant differences in the contents of alcohols, aldehydes, and ketones were observed. A relative odor activity value (ROVA) analysis of the aroma compounds revealed that 19 compounds had an ROVA > 1. Among them, benzylaldehyde, trans-2-decenal, decanal, benzaldehyde, nonanal, hexanal, trans-linalool, and geraniol from the shaking withering method had significantly higher ROVA values than those from the other withering methods, which may be the reason for the prominent floral and fruity aroma of shaking withering. This study revealed the impact of different withering methods on the quality of Congou black tea, providing a scientific basis for the development of Congou black tea with different flavors and the improvement of Congou black tea processing techniques.

## 1. Introduction

Black tea has a bright red color and a rich and mellow taste, and is known for its various health benefits such as antioxidant properties, blood sugar and blood pressure reduction, and weight loss [1]. It is loved by consumers worldwide because it is suitable for mixing with other ingredients [2]. Black tea can be classified into souchong black tea, Congou black tea, and broken black tea on the basis of the different processing methods used [3]. The processing technologies of Congou black tea, the mainstream product of Chinese black tea, includes withering, rolling, fermentation, and drying [4,5]. In the processing of Congou black tea, withering is crucial for its quality [6]. It is not only the basis for the smooth progress of subsequent processes, but also a prerequisite for the quality of black tea [7]. Therefore, research on withering methods is necessary [8].

Withering is the first step in the production of Congou black tea, which involves controlling the rate and degree of moisture loss in fresh tea leaves by adjusting the temperature, humidity, and lighting conditions [9]. The wilting process causes the color of the leaves to darken, water loss leads to softening of the leaf tissue [10], and enzyme activity changes. Additionally, self-decomposition of internal substances occurs, altering the aroma [11]. Research has shown that different withering methods can affect the composition and components of black tea, which in turn affects its quality [12]. Currently, many new withering methods such as freezing withering [13], red light withering [14], dynamic withering [15], and yellow light withering [16] are being researched. However, the most commonly used methods in actual production are still natural withering and warm-air withering [17]. Sunlight withering has limitations in terms of production because of the influence of climate factors in different regions. In recent years, the key processing technique “shaking”, which imparts floral and fruity aromas to oolong tea, has also been introduced into the withering process of Congou black tea to enhance its fragrance [18]. At present, there is a lack of systematic and comprehensive comparative studies on withering methods. In this study, five different withering methods, including natural withering (NAW), warm-air withering (WAW), sun–natural combined withering (SNW), sun withering (SUW), and shaking withering (SHW), were used to produce Congou black tea, and a comprehensive analysis was conducted to explore the comprehensive effects of different withering methods on the quality of Congou black tea. This study is highly important for the optimization and innovation of Congou black tea processing, breaking the homogenization of Congou black tea, and improving the quality of Congou black tea.

## 2. Materials and Methods

### 2.1. Tea Sample Preparation

Fresh tea leaves (cultivar “yunkang No. 10”) of one bud and two leaves were obtained from Xiaoguan Tea Industry Company in Fengqing County, Yunnan Province, and the processing method was carried out as follows: withering, rolling, fermenting, and drying. Each withering method was repeated three times, with 12 kg of fresh leaves used for each method. Finally, the prepared samples were stored in a refrigerator at −20 °C.

For the natural withering treatment (NAW), the leaf thickness was 3 cm, temperature 21–24 °C, humidity around 65%, and wilting time 12 h; for the warm-air withering treatment (WAW) the leaf thickness was 15 cm, warm-air temperature 24 °C, humidity 59%, and withering time 9 h. For the sun–natural combined withering (SNW) treatment, the temperature was 21–30 °C, the humidity was 45–65%, and the withering time was 8 h. For the sun withering treatment (SUW), the leaf thickness was 2 cm, the temperature was 26–30 °C, the humidity was 45–50%, the withering time was 2.5 h, and it needed to be completed before 11 a.m. For the shaking withering treatment (SHW), the fresh tea leaves were shaken four times in total. The parameters for the first round of shaking were as follows: 6 r/min for 2 min, then the mixture was spread out for 45 min. The second round of shaking was 10 r/min for 3 min and spread out for 60 min. The third round of shaking was 13 r/min for 2 min and spread out for 90 min. The fourth round of shaking was 13 r/min for 5 min. The leaf thickness was 3 cm, the temperature was 23–25 °C, the humidity was around 65%, and the withering time was 9 h.

The withered leaves were then placed into the digital rolling machine. The mixture was subjected to 10 min of empty rolling, followed by 25 min of light-pressure rolling. Then, the samples were rolled heavily for 15 min, and subjected to 10 min of empty rolling. The total duration was 60 min. After unblocking, the leaves were transferred to a fermentation box for fermentation.

The fermentation cabinet was then moved into the fermentation room at a constant temperature of 25–28 °C and a relative humidity of 95% for fermentation. The fermentation duration was 4.5 h.

After fermentation, the fermented leaves were dried at 110 °C for 20 min until the moisture content reached 20–25%, cooled for 30 min, and then, heated at 85 °C until the moisture content reached approximately 6%.

### 2.2. Chemicals

Sodium chloride, ethyl acetate, ethyl alcohol, oxalic acid, methanol, sodium bicarbonate, anhydrous sodium carbonate, basic lead acetate, stannous chloride, hydrochloric acid, normal butanol, xanthone, potassium dihydrogen phosphate, concentrated sulfuric acid, and anthraquinone were purchased from Damao Chemical Reagent Factory (Tianjin China). Folin-phenol (biochemical reagent), phosphoric acid, and acetonitrile, were purchased from Beijing Solarbio Science & Technology Co., Ltd. (Beijing China). Ethyl decanoate (≥99% purity grade) was purchased from Aladdin Biological Co., Ltd. (Shanghai, China). Sykam original pH-type amino acid standard solution, sample dilution solution, and elution buffers with different pH and ion strengths (SKYAM (Beijing) Scientific Instrument Co., Ltd., Beijing, China); catechin, epicatechin, epigallocatechin, gallocatechin gallate, and epigallocatechin gallate were purchased from Lemeitian Pharmaceutical Technology Co., Ltd. (Chengdu, China).

### 2.3. Instruments and Equipment

A 6CWD-16 W tea withering machine, and 6CFJ-100 tea fermentation machine (Changsha XiangFeng Intelligent Equipment Co., Ltd., Changsha, China) were used. A ZCR35-type digital twisting machine (Jinzhou Huihang Pharmaceutical Machinery, Jinzhou, China) was used.

A UV-2102PC UV‒vis spectrophotometer (Element Analytical Instrument Co., Ltd., Shanghai, China), 1525 HPLC-type high-performance liquid chromatograph (WATERS, Milford, MA, USA), 2487 dual-absorbance UV detector (λ = 278 nm, WATERS, USA), C18 column (250 × 4.6 mm, 5 µm, Agilent, Santa Clara, CA, USA); amino acid analyzer (SYKAM, GE), and QP2020NX gas chromatography‒mass spectrometry system (Shimadzu Corporation, Kyoto City, Japan) were used. The sample separation was performed using an Rtx-5MS chromatographic column (60 m × 0.25 mm × 0.50 µm, Shimadzu Corporation, Japan), B13-3 intelligent constant-temperature timing magnetic stirrer (Sile Instrument Co, Ltd., Shanghai, China), and 50/30 μm DVB/CAR/PDMS solid-phase microextraction fiber (Supelco, Bellefonte, PA, USA).

### 2.4. Sensory Evaluation

The sensory characteristics of 15 samples of Congou black tea subjected to 5 different withering treatments were evaluated according to GB/T 23776-2018 “Methodology for sensory evaluation of tea” of China. The tea tasting panel was composed of 7 experienced experts (4 females, 3 males, aged 30–50) divided into 3 groups. A mass of 3 g of dried tea leaves was weighed into a white porcelain cup (150 mL) specially designed for sensory evaluation, then 100 °C hot water was poured in and it was steeped for 5 min before pouring out the tea infusion. The appearance, liquor color, aroma, taste, and infused leaves of the samples were evaluated. The final scores were the average of the scores given by the 3 groups of experts, and the results are presented as the mean ± standard deviation.

### 2.5. Determination of Nonvolatile Components

The detection methods for caffeine, theaflavins, free amino acids, and soluble sugars are consistent with the methods published by Yang et al. [19]. The analysis of amino acid and catechin components was performed at the Yunnan Academy of Agricultural Sciences Tea Research Institute. The catechin component content referred to GB/T 8313-2018. The gradient elution method was used for determination, in which the mobile phase A was an aqueous solution containing 0.02% acetic acid, 0.09% acetonitrile, and 0.002% ethylenediaminetetraacetic acid disodium salt (EDTA-2Na). The mobile phase B was an aqueous solution containing 80% acetonitrile, 0.02% acetic acid, and 0.002% ethylenediaminetetraacetic acid disodium salt (EDTA-2Na). The flow rate was 1 mL/min and the column temperature was 35 °C. The amino acid component content referred to GB/T 30987-2020 for determination.

### 2.6. Determination of Volatile Components

#### 2.6.1. Extracting Components of Tea Aroma

HS-SPME was used to extract aromatic components from tea leaves via headspace solid-phase microextraction. One gram of tea sample was accurately weighed and added to a 20 mL solid-phase microextraction headspace vial. Then, 1 g of sodium chloride was added to saturate the solution, followed by the addition of 5 mL of distilled water at 60 °C and 5 µL of 50 µg/mL ethyl cinnamate. The headspace extraction vial was placed on a magnetic stirring hot plate and heated at a constant temperature of 60 °C. After stabilizing for 6 min, the SPME fiber (50/30 um DVB/CAR/PDMS, Supelco, Bellefonte, PA, USA) was inserted into the headspace vial for aroma adsorption for 50 min. Then, the fibers were removed and introduced into the GC‒MS instrument for desorption and analysis.

#### 2.6.2. Analysis Conditions and Qualitative and Quantitative Methods of Tea Aroma Components

The aroma components of the experimental tea samples were analyzed by the QP2020NX gas chromatography‒mass spectrometry system (GC‒MS). The specific analysis conditions for the gas chromatograph and mass spectrometer were as follows. The GC conditions were as follows. Rtx-5MS column (30 m × 0.25 mm × 0.25 μm): initial column temperature of 40 °C, no hold, ramped at 1.5 °C/min to 70 °C, no hold, ramped at 1 °C/min to 72 °C, held for 2 min, ramped at 1.5 °C/min to 90 °C, held for 1 min, ramped at 5 °C/min to 170 °C, held for 1 min, finally, ramped at 9 °C/min to 230 °C, held for 2 min; injection port temperature set at 250 °C; split mode; carrier gas helium, column flow rate of 1.0 mL/min. MS conditions: EI source, 70 eV; ion source temperature, 230 °C; interface temperature, 240 °C; mass scanning range, *m*/*z* 40∼400; and solvent delay time 2 min.

All volatile compounds were initially identified via the National Institute of Standards and Technology (NIST 17) mass spectrometry library. Each compound was further confirmed through retention index (RI) matching. The RI value was calculated via linear interpolation on the basis of the retention time of the n-alkanes. Quantitative analysis of volatile compounds was performed on the basis of the peak areas of internal standard compounds.

### 2.7. Data Analysis

The Excel 2021 software was used to organize the data, conduct standard deviation analysis for sensory evaluation scores (±), and create a bar graph; the SPSS 26.0 software was used for analysis of variance and the least significant difference test, and the significance of the compound concentrations was calculated (*p* < 0.05); and chiplots and TBtools were used for drawing grouped error line charts, within-group correlation heatmaps, aroma substance heatmaps, and stacked bar graphs.

## 3. Results

### 3.1. Impact of Different Withering Methods on Sensory Evaluation

The different withering methods have a certain impact on the appearance, liquor color, aroma, and taste of Gongou black tea, and their scores and differences are shown in Table 1. The total scores of SHW and NAW differed significantly from those of the other four withering methods (*p* < 0.05), and the highest total sensory scores were obtained for SHW (95.2 ± 0.72), followed by NAW (93.7 ± 0.34). There was no significant difference in the total scores for SUW, SNW, and WAW (92.7 ± 0.08, 92.5 ± 0.37, 92.5 ± 0.60). The aroma scores of SHW differed significantly from those of the other four wilting methods (*p* < 0.05) and had the best aromas, with obvious floral and fruity aromas, accompanied by milky aromas, rich and long-lasting aromas, and the cold aroma still had a floral aroma; WAW and SNW had floral aromas, but they were weaker than those of SHW and less persistent than those of SHW. The taste scores of SHW and NAW were not significantly different from each other but were significantly different from those of the other three types of withering methods (*p* < 0.05), with SHW and NAW being more balanced in terms of taste and showing a rich, mellow, and refreshing flavor, whereas SUW and SNW had a stronger and more intense taste, with SUW having a bitter taste, SNW having a slightly astringent taste, and WAW having a lighter and slightly astringent taste compared with the other four types of withering methods.

### 3.2. Impact of the Five Withering Methods on the Biochemical Components of Congou Black Tea

Five different withering methods and their biochemical components are shown in Figure 1. Tea polyphenols is the general term for phenolic substances in tea leaves, and they are the key components with health benefits in tea [20]. Owing to their high content in water extracts, they have an important influence on the strength of tea infusion. The content of tea polyphenols in SNW was the highest at 17.22%, while the content in WAW was the lowest at 12.94%. There was no significant difference between SNW and SUW, but significant differences were observed among the other withering methods (*p* < 0.05). Free amino acids are not only important factors in the quality of tea but also have excellent nutritional and medicinal functions. There are significant differences in the free amino acid content (*p* < 0.05), with the highest content in the WAW treatment (3.67%) and the lowest in the SUW treatment (3.42%). Caffeine is an important characteristic substance in tea, with effects such as being refreshing, diuretic, and metabolism promotion. It mainly presents a bitter taste and can form hydrogen bonds with catechins to increase the freshness of the tea infusion and reduce its bitterness. Among the five types of withering methods, the WAM method had the highest caffeine content (5.16%), which was significantly different from that of the SUW and NAW methods (*p* < 0.05). This may be related to the fresh and mellow taste of tea brewed via the WAM method. The total soluble sugars not only give the tea infusion a sweet and mellow taste [21] but are also related to the “caramel aroma” of the tea leaves. Among the five withering methods, there was a significant difference in the content of soluble sugars (*p* < 0.05), with the highest content of 3.90% in the SUW treatment and the lowest content of 3.32% in the SHW treatment.

### 3.3. Effects of Different Withering Methods on Amino Acid Composition and Content

Amino acids can affect not only the flavor quality of tea but also the aroma of tea [22]; if phenylalanine is present, tyrosine is bitter in taste, and at the same time, it is an aromatic amino acid. During the withering process of black tea, proteins and peptides in fresh leaves undergo self-dissolution and decomposition. The speed and extent of this decomposition are related to the withering method, which results in varying compositions and proportions of amino acids, ultimately impacting the overall quality of the tea. A total of 22 amino acids (Figure 2) were detected in the five withering methods, and there was no significant difference in the amino acid species of each withering method, with the highest number of 22 species in the WAW and the lowest number of 20 species in the SHW. All the amino acids were analyzed by clustering and the results are shown in Figure 2, where all the wilting methods are classified into two main groups, with the SUW treatment and SNW treatment as one group and the other withering methods as one group. There was no significant difference in the content of bitter amino acids. The process with the lowest sweet amino acid content (Figure 3A) was SHW at 1.40 mg/g, while the highest content was found in SNW at 1.79 mg/g. Among all the sweet amino acids, serine (Ser) had the highest content, with the SHW serine content (Figure 3B) being the lowest at 0.45 mg/g and the SUW content being the highest at 0.60 mg/g. There were significant differences in the total content of sweet amino acids and serine content among SHW, SNW, and SUW (*p* < 0.05). The content of umami amino acids (Figure 3C) was lowest in SHW at 11.38 mg/g and highest in SNW at 12.70 mg/g, with a significant difference (*p* < 0.05) between the two. A comparative analysis of the contents of all the umami amino acids revealed that the theanine content was not significantly different, whereas the glutamic acid (Glu) and aspartic acid (Asp) contents (Figure 3D) were significantly different (*p* < 0.05), with the highest content of glutamic acid in SNW (1.52 mg/g) and the highest content of aspartic acid in NAW (0.65 mg/g). Among the five types of withering methods, the WAW amino acid type is the most abundant, whereas SNW and SUW are more conducive to the accumulation of sweet amino acids. The NAW and SNW treatments were more conducive to the accumulation of umami amino acids, whereas the SHW treatment resulted in lower levels of all the taste amino acids. Huang et al. [23] reported that warm-air withering and sun withering can significantly reduce the loss of nonprotein amino acids (theanine). In this study, there was no significant difference in the theanine content among the five withering methods, and theanine content was highest in NAW, followed by WAW and SNW. This may be related to the different tea tree varieties and environmental parameters used during processing.

### 3.4. Effects of Different Withering Methods on the Composition of Catechins

Catechins are important polyphenolic compounds found in tea leaves, and their individual composition and content significantly affect the taste and quality of tea. During the processing of black tea, catechins serve as important conversion substances for the formation of theaflavins and thearubigins, which are closely related to the color and taste of black tea. The research results of Sanderson et al. [24] showed that epicatechin and epigallocatechin only have a bitter taste, without astringency, whereas gallocatechin gallate and epigallocatechin gallate have both bitter and astringent tastes. Scharbert et al. [25] used inhalation–exhalation techniques to demonstrate that ester-type catechins in tea are the main substances responsible for the astringency of tea, with epigallocatechin gallate in particular contributing the most. As the main components of tea infusion flavor substances, various catechins vary in composition and content, resulting in different flavors of tea infusion. In this study, catechin (C), epicatechin (EC), epigallocatechin (EGC), epigallocatechin gallate (ECG), and epigallocatechin gallate (EGCG) were detected. The contents of the catechin components obtained via the five withering methods are shown in Figure 3. Among them, in the SNW, the content of C was the highest at 0.179%, which was significantly different from those of the other withering methods. In the NAW treatment, the content of EC was the highest at 0.25%, which was significantly different from those in the other three withering treatments, except for WAW (*p* < 0.05). The EGC content in NAW was highest at 0.55% and significantly different from that in SHW (*p* < 0.05). ECG and EGCG had the lowest contents in SHW at 1.00% and 0.48%, respectively, and were significantly different from those in the other four treatment methods (*p* < 0.05). The ECG and EGCG contents were highest in SNW at 2.04% and 1.48%, respectively. Among the five withering methods, SHW had the lowest content of all the catechin fractions, which may be related to the oxidation of polyphenolic substances during the shaking process, followed by NAW, and SNW had the highest content of all the fractions. Combining Figure 1A with Figure 4, the lower tea polyphenol content but higher ester catechin content might be the reason for the lighter flavor but astringent taste of the WAW treatment.

### 3.5. Effects of Different Withering Methods on Tea Pigments and Correlation Analysis with Other Nonvolatile Substances

Theaflavins (TFs), thearubigins (TRs), and theabrownins (TBs) are not only key substances that contribute to the color of infusion and infused leaves [26], but also strongly correlate with the flavor of tea. Additionally, they are important bioactive compounds in black tea [27]. TF are the main components that impart a bright red color to teainfusion, and they also affect the concentration, strength, and freshness of the flavor of black tea [28]. TRs constitute the main component that gives black tea its red color, with a certain level of astringency and a sweet and mellow taste. TBs cause the tea liquor to appear dark brown and taste bland, which is the main reason for the dark color of black tea. Therefore, in the processing of black tea, it is advisable to preserve more TFs and TRs while reducing the accumulation of TBs. Among the five withering methods, there was no significant difference in the contents of TFs (Figure 5A), but SHW had the highest content at 0.24%. There were significant differences in the contents of TRs and TBs (Figure 5B) (*p* < 0.05), with SHW having the highest TR content at 8.69% and SNW having the lowest at 6.50%. SHW also had the highest TB content at 8.25%, while WAW had the lowest at 5.230%. The total ratio of TFs and TRs to TBs is an important factor in assessing the quality of black tea. As shown in Figure 5C, the SHW treatment had the lowest ratio (1.08) and was significantly different from the other four withering methods (*p* < 0.05), possibly because enzymatic oxidation started at the edges of the fresh leaves during withering. The withering method with the highest ratio was the WAW treatment at 1.44. Figure 5D shows that the trends in the contents of TFs, TRs, and TBs among the five withering methods are consistent, but opposite to those of the catechins. The results of the intragroup correlation analysis of the nonvolatile substances from the 15 groups subjected to the five withering methods are shown in Figure 6, indicating that the TR, TB, soluble sugar, caffeine, free amino acid, tea polyphenol, and catechin fractions presented different degrees of negative correlation. The catechin fractions, on the other hand, were essentially positively correlated with tea polyphenols, soluble sugars, caffeine, and free amino acids to varying degrees. This may be related to the coupled oxidation of catechin o-quinone.

### 3.6. Analysis of Volatile Substances Under Different Wilting Methods

Aromatic substances, as the main volatile substances in black tea, are important indicators of sensory evaluation and have a significant impact on the overall quality and market value of black tea [29]. A total of 227 aroma components, including 58 alcohols, 36 aldehydes, 23 ketones, 59 acid esters, 42 hydrocarbons, and 9 other substances, were detected via the five different withering methods for black tea, and the percentage of each component is shown in Figure 7. Among them, alcohols accounted for 62–66% of the total volatile components, and the alcohol content was highest in SNW and differed significantly from that in SHW and NAW (*p* < 0.05). The linalool content of linalool was the highest among all alcohols, indicating that regardless of the type of withering, the main aroma components of Dianhong black tea did not change [30]; however, the linalool content differed among the different methods of withering, with the lowest content being 33% in SHW, which was significantly different from those of the other four withering methods (*p* < 0.05). There was no significant difference in the relative contents of acid esters, hydrocarbons, or other substances. The aroma of tea is usually determined by volatile compounds (VCs), and the contribution of VCs to the aroma is usually evaluated by the relative odor activity value (ROAV) [31,32]. Usually, an aroma substance with an ROAV ≥ 1 is identified as the active aroma compound in tea, and 0.1 ≤ ROAV < 1 is considered to be an important modifier of the overall flavor of tea. Since the ROAV value corresponds to the magnitude of the contribution of volatile components to aroma, to further analyze the effects of different withering methods on the aroma of Congou black tea, the ROAV values were calculated for the aroma substances separately, and all the aroma substances with ROAV > 0.1 were subjected to clustering heatmap analysis (Figure 8). The results revealed that the SHW treatments were clustered separately and that all had higher ROAV values than the other treatments did. Among the 15 samples subjected to the five different withering methods, a total of 19 aroma compounds with ROAV ≥ 1 were detected. A differential analysis was conducted on the 32 compounds with ROAV ≥ 0.1, and 19 compounds were significantly different (*p* < 0.05) between the treatments. The presence of benzylaldehyde, trans-2-decenal, decanal, benzaldehyde, nonanal, hexanal, trans-linalool, geraniol, and 1-octen-3-one, in combination with ROAV values, may explain the pronounced aroma in the SHW treatment compared with the other treatments.

## 4. Discussion

This study conducted a comprehensive and systematic analysis of five different withering methods for Congou black tea, revealing the impact of different withering methods on its internal components. In previous studies, Huang et al. [33] compared the effects of sun withering, natural withering, and warm-air withering on the quality of Qimen black tea. The research results revealed that sun-withered tea had high levels of polyphenols and soluble sugars, but low levels of free amino acids, which is consistent with the results of this study. Sun withering is the process in which fresh leaves lose water under sunlight, during which both the respiration of fresh leaves occurs and also intense photosynthesis takes place. Research by Zeng et al. [34] has shown that sunlight withering promotes carbon metabolism in withered leaves and inhibits nitrogen metabolism. In withered tea leaves subjected to sunlight, the synthesis and breakdown of catechins, as well as the synthesis and consumption of soluble sugars, are quite intense. These findings may explain the high content of catechins and soluble sugars in tea leaves subjected to sunlight withering and the low content of free amino acids. Although the tea polyphenols may be low, their individual catechin components are high. Perhaps this is because warm-air withering significantly upregulates differentially expressed genes (DEGs) related to catechin, resulting in an increase in catechin concentration [23]. Among the five withering treatments, SHW had the highest contents of aldehyde and ketone aroma compounds, including benzaldehyde, phenylacetaldehyde, nonanal, furfural, 2-heptanone, 3-undecen-2-one, and acetophenone, which were significantly different from those of the other four treatments. SHW had the lowest content of linalool, while the content of linalool isomers such as geraniol was the highest and significantly different from those of the other four treatments. Shaking promotes cell rupture, allowing the precursors of aroma compounds to mix with enzymes and undergo enzymatic reactions. It is also associated with the oxidation of catechins during the process. The coupled oxidation of quinones promotes the formation of aldehydes, ketones, and the oxidation of linalool, which is consistent with the findings of Zeng et al. [34] and Yin et al. [35]. Compared to other withering methods, natural withering with long and moderate water loss is conducive to the full transformation of internal components. It has higher levels of free amino acids and soluble sugars. Although it has higher levels of tea polyphenols, the bitterness of caffeine is the lowest. It also has a higher ratio of TFs and TRs to TBs, which may be the reason why NAW treatment has superior sensory evaluation. Owing to the fact that this study only examined dried samples and lacked a dynamic study on the effects of different withering methods on the quality of Congou black tea, the next step of this study will involve the use of metabolomics techniques to analyze fresh tea leaves, withered tea leaves, and dried tea to explore the dynamic changes in the components of Congou black tea resulting from different withering methods. It is hoped that through the above research, necessary data support will be provided for the optimization and adjustment of the processing parameters of Congou black tea in the future.

## 5. Conclusions

This study sampled five types of Congou black tea processed with different withering methods for sensory evaluation and volatile and nonvolatile compound analysis to explore the effects of different withering processes on the taste and aroma characteristics of Congou black tea. The results showed that the withering method could affect the contents of tea polyphenols, catechin fractions, soluble sugars, free amino acids, caffeine, thearubigins, theabrownins, and aroma substances of Congou black tea, which presented different concentrations, thus affecting the quality of Congou black tea, and there were different degrees of correlation among the internal components. Natural withering is more balanced in terms of quality factors; however, it has the issue of low efficiency; warm-air withering has a low polyphenol content but is high in ester-type catechins, the taste has a slightly astringent sensation, but its sweetness and fragrance are excellent, and its parameters are controllable, ensuring a stable quality of the produced black tea. Shaking withering has a strong effect on the aroma of Congou black tea and reduces the bitterness and astringency of the tea; therefore, it may be possible to improve the quality of summer tea; both sun withering and sun–natural combined withering are high in polyphenol content and have strong flavors, due to the climatic conditions in Yunnan, these two withering methods still face certain difficulties in production. In this study, the effects of different withering methods on the quality of Congou black tea were analyzed at the level of volatile and nonvolatile compounds, which provided some theoretical basis for the mechanism of Congou black tea quality formation, quality enhancement, and innovation of process integration in the later stages.

## Figures and Tables

**Figure 1 foods-13-03456-f001:**
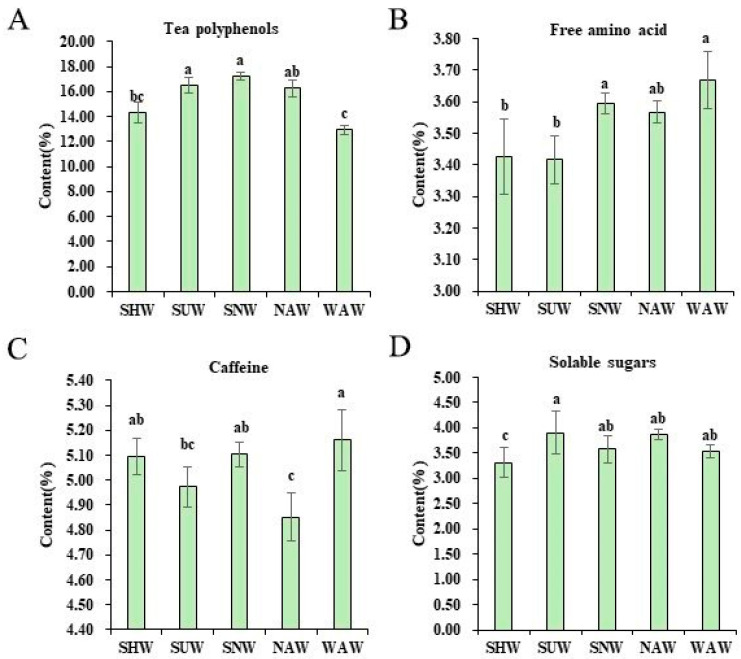
The content of tea polyphenols (**A**), free amino acids (**B**), caffeine (**C**), and soluble sugars (**D**) in Congou black tea subjected to different withering methods (%); different letters in the same subgraph indicate significant differences (*p* < 0.05).

**Figure 2 foods-13-03456-f002:**
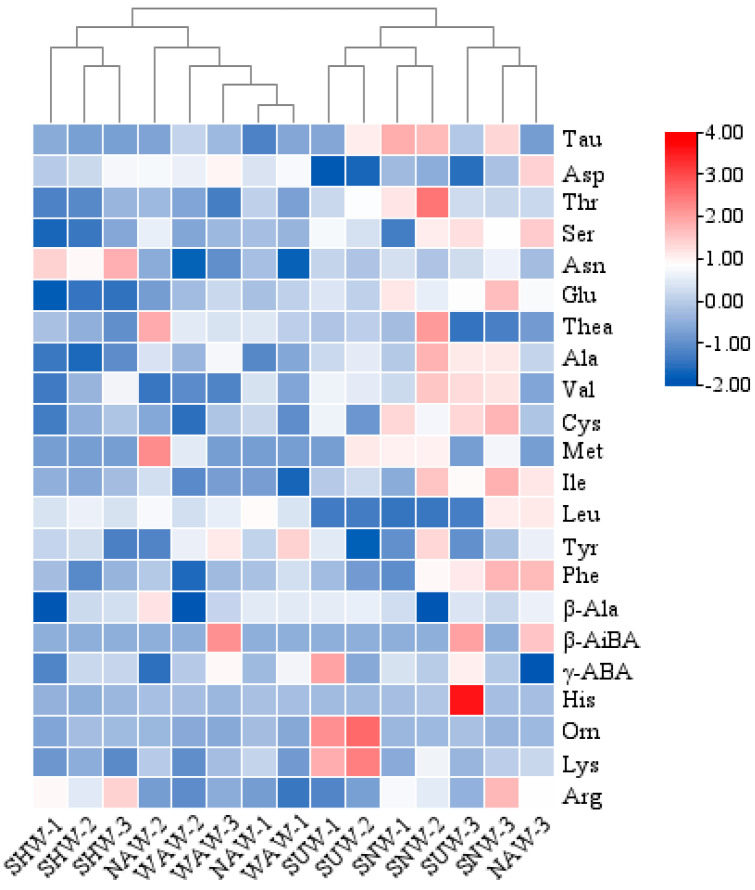
Clustering heatmap of amino acid fractions obtained via different withering methods.

**Figure 3 foods-13-03456-f003:**
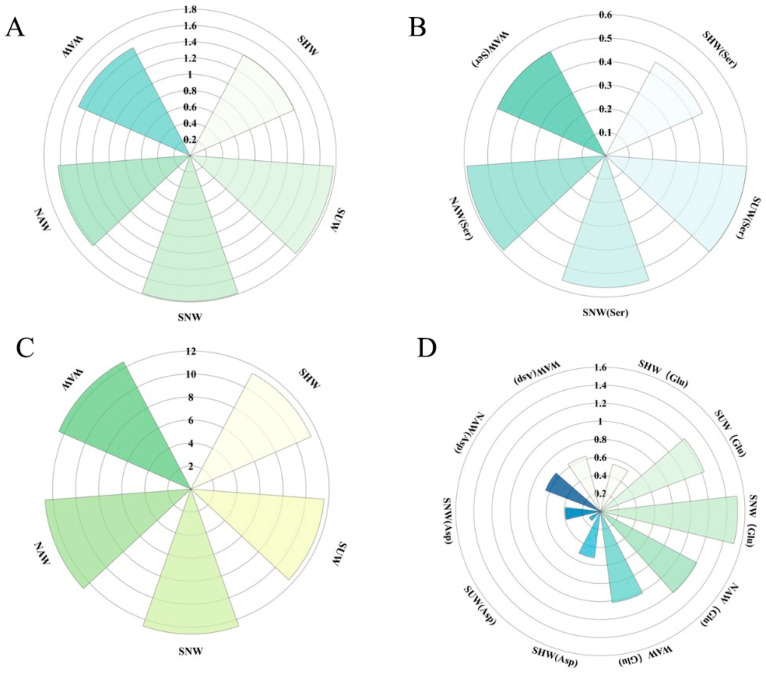
Rose diagram of the sweet amino acid content (**A**), serine content (**B**), umami amino acid content (**C**), and glutamic acid and aspartic acid contents (**D**) under different withering methods; natural withering (NAW), warm-air withering (WAW), sun–natural combined withering (SNW), sun withering (SUW), shaking withering(SHW), serine (Ser), glutamic acid (Glu), aspartic acid (Asp).

**Figure 4 foods-13-03456-f004:**
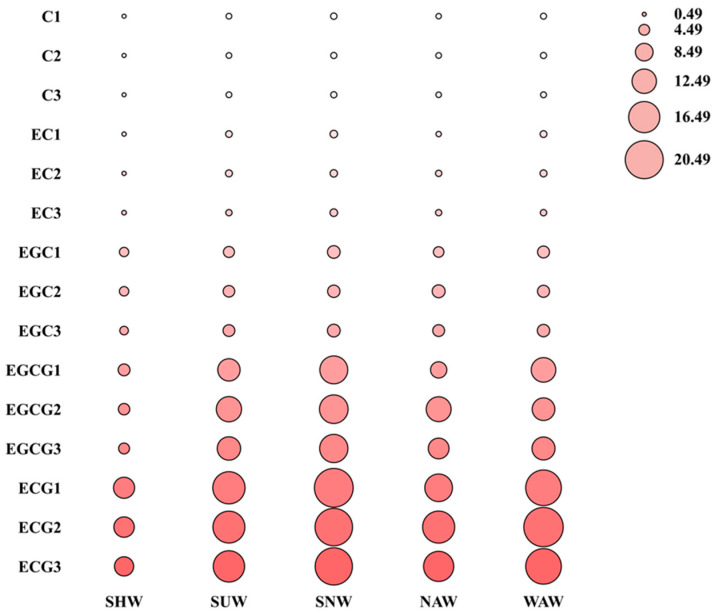
Bubble diagram of the catechin compositions of different withering methods.

**Figure 5 foods-13-03456-f005:**
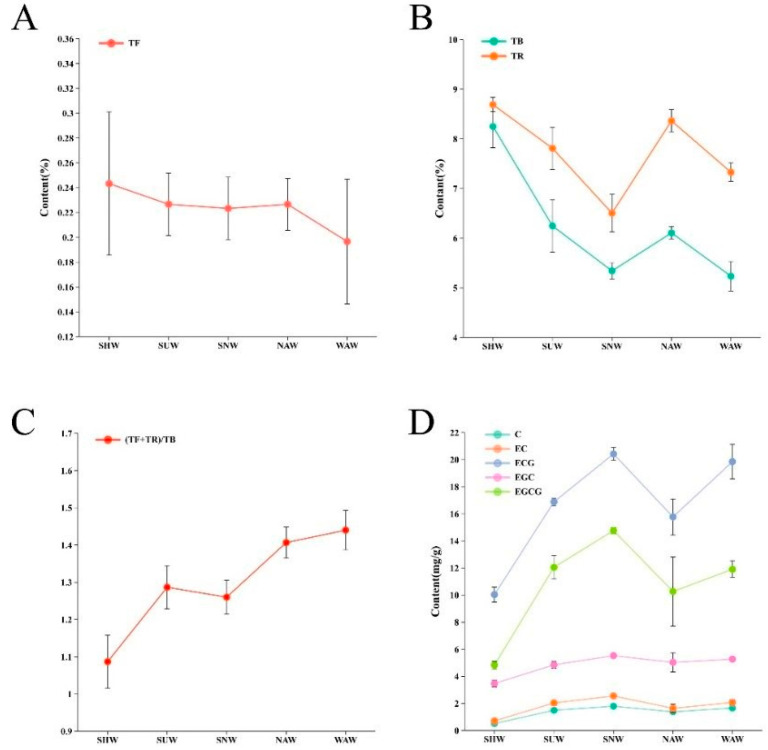
Group error line graphs of catechin (**A**), line graph of the grouping error of theophylline and theobromine (**B**), line graph of the grouping error of the ratio of theaflavins and thearubigins to theabrownins (**C**), line graph of group error for tea catechin components (**D**).

**Figure 6 foods-13-03456-f006:**
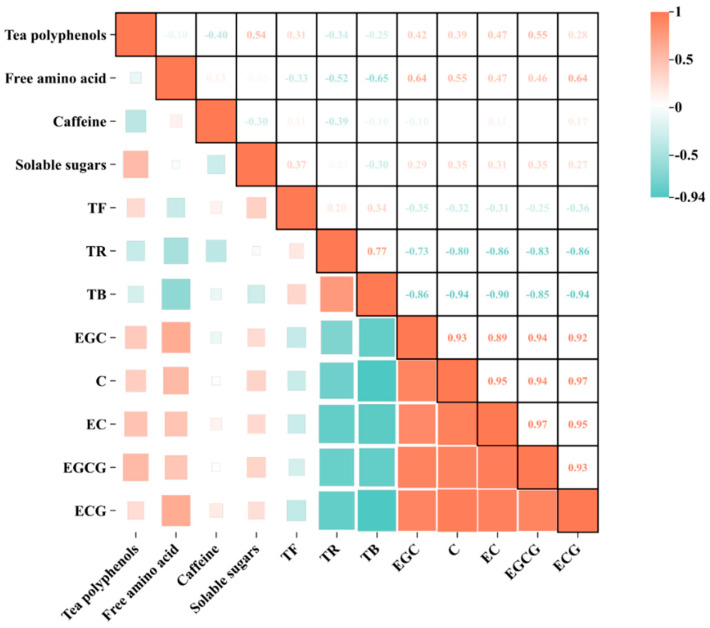
Heatmap of intragroup correlation analysis of nonvolatile inclusions under different withering methods.

**Figure 7 foods-13-03456-f007:**
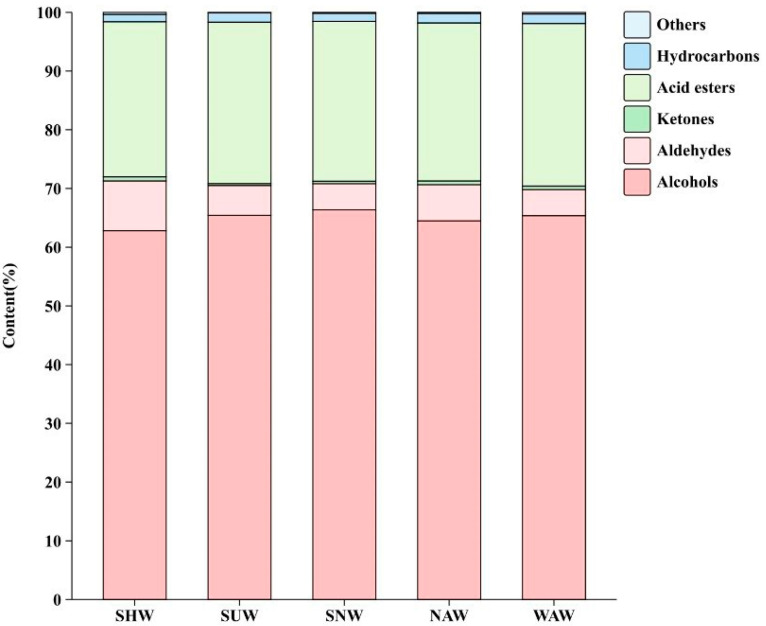
Types of aroma substances and their proportions under different withering methods.

**Figure 8 foods-13-03456-f008:**
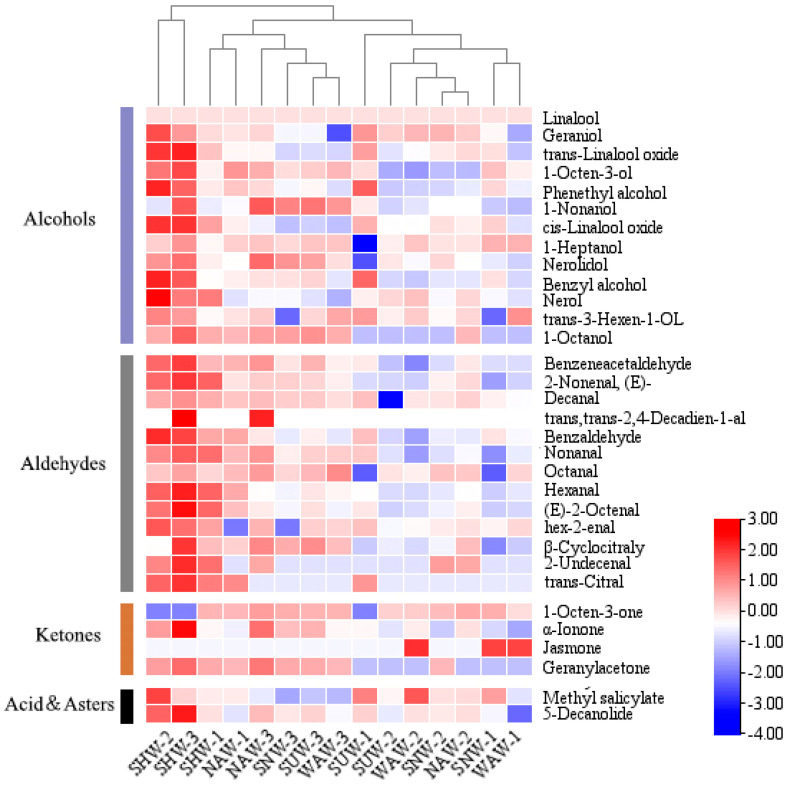
Clustering heatmap analysis of all aroma substances with ROAV > 0.1 for different withering methods.

**Table 1 foods-13-03456-t001:** Sensory evaluation results of black tea with different withering methods.

Sample	Appearance and Score (25%)	Liquor Color and Score (10%)	Aroma and Score (25%)	Taste and Score (30%)	Infused Leaf and Score (10%)	Total Score
SHW	The color of the tea is dark brown with good luster and golden hairs 95.0 ± 0.00 a	Red ^(+)^, bright 95.0 ± 0.00 a	Floral and fruity, with a strong and persistent aroma 95.1 ± 2.00 a	Mellow, smooth, fresh 95.9 ± 0.92 a	Uniformly red in color and glossy 94.3 ± 1.15 a	95.2 ± 0.72 a
SUW	The color of the tea is dark brown with luster 94.0 ± 0.00 ab	Orange–red, bright ^(+)^ 94.3 ± 0.66 ab	Sweet aroma, persistent ^(˗)^ 91.4 ± 0.42 b	Strong, slightly bitterness 92.2 ± 0.17 b	Uniformly red in color ^(˗)^, glossy ^(˗)^ 93.7 ± 1.15 a	92.7 ± 0.08 c
SNW	The color of the tea is dark brown with more golden hairs 94.0 ± 1.00 ab	Red–orange, bright 93.0 ± 1.00 bc	Sweet, floral ^(˗)^, and persistent aroma 92.3 ± 1.00 b	Strong, astringent 91.4 ± 0.35 b	Uniformly red in color 93.0 ± 0.00 a	92.5 ± 0.37 c
NAW	The color of the tea is dark brown with luster and fewer golden hairs 93.3 ± 0.58 b	Orange–red ^(+)^, bright ^(+)^ 94.6 ± 0.53 a	Sweet aroma, persistent ^(+)^ 91.8 ± 0.59 b	Strong, fresh 95.7 ± 0.10 a	Uniformly red in color ^(˗)^, glossy ^(˗)^ 93.7 ± 1.15 a	93.7 ± 0.34 b
WAW	The color of the tea is dark brown with golden hairs 93.0 ± 0.00 b	Red–orange ^(˗)^, bright ^(+)^ 92.6 ± 0.92 c	Sweet, floral ^(+)^, and persistent aroma 92.4 ± 1.50 b	Mellow, fresh slightly astringent 95.9 ± 0.92 b	Uniformly red in color ^(˗)^ 92.3 ± 0.58 a	92.5 ± 0.60 c

Notes. The total score is 100 points: 25% represents the proportion of dry tea appearance; 10% represents the proportion of liquor color; 25% represents the proportion of aroma; 30% represents the proportion of taste; 10% represents the proportion of infused leaves. (˗) means that this characteristic is present but slightly less intense; (+) means that this characteristic is present and slightly more intense; different letters for the same factor indicate a significant difference between groups (*p* < 0.05).

## Data Availability

The original contributions presented in the study are included in the article, further inquiries can be directed to the corresponding authors.

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
