# Peer review of "Effects of Five Different Withering Methods on the Composition and Quality of Congou Black Tea"

_foods, 2024, doi:10.3390/foods13213456_

Round 1
Reviewer 1 Report
Comments and Suggestions for Authors
Dear Authors,
the work subject is very interesting and sound scientifically. However some points can be improved to become more interesting and clear to the readers.
1 – Graphical abstract
I suggest to increase the figures size. they are too small.
Material and methos
2- In order to be comparative, the leaf thickness size shouldnt be the same for all tested methods? I think the thickness will directly impact in area surface and contact and the dryness speed.
3 - what was the leaf thickness in the SHW? and the humidity of 65% was lasted during the whole process
4 - Why you add 1g of sodium chloride and ethyl cinnamate in the headspace? Wht it means as bennefits to the sample extraction process?
5 – Why have u chosen the DVB/CAR/PDMS fiber? Why not PDMS or DVB/CAR? Have u tested other fibers before to check how they work on your products?
6 - Qualitative analysis was done using GC-MS or GC-FID?
Results:
1- Is there any information about the concentration of phenolic compounds befre and after the withering process?
2- Why shaking process has the loewst and the SNW and WaW the highest polyphenolic content? what has impact the oxidation?
3- the information in this Figure is too pale light color, it is not good quality. I suggest to improve it.
4- I really missed the chromatograms to compare the chemical profile. I think it is important to understand visually the impact of the different withering methods on chemical profile.
5- Very interesting, I really thought that WAW could negatively impact the presence of monoterpene derivative compounds in the oils due to their higher volatility, how the essential oils are stored in the leaves? It seems they are well protected.
Discussion
I think the discussion can be improved. I would expect some data about advantages and disadvantages of each method used taking different aspects in consideration like flavors (taste, aroma); colors; antioxidant properties, and loss of nutritional components by accelerate oxidation due to some withering process.
It is known that natural as well as some indoor withering methods tend to preserve higher levels of polyphenols, since in this process the leaves gradually lose moisture, while enzymes like polyphenol oxidase remain active. This fact is described to contribute to the tea’s astringency and antioxidant properties, which is interesting in some cases. On the other hand, outdoor and combined withering methods, could enhance the production of volatile aromatic compounds, giving the tea a stronger fragrance and richer flavor.
In some cases, like large amounts in industry, the mechanical hot-air withering is advantageous for mass production due to its efficiency, however it is reported to also compromise the nuanced qualities of the tea, since the high temperatures accelerate enzymatic reactions and moisture evaporation, it can reduce the complexity of the tea's aroma and lead to a loss of certain more volatile compounds, like monoterpene derivatives that can contribute to the subtlety of Congou black tea.
Due to this I would suggest to insert the HPLC chromatogram in the files to really see before and after how the withering process impact the chemical constituents in the tea composition. Also CG-MS of SPME to check about the aroma would be interesting.
Conclusion
Can be improved
Ex: “The results showed that the withering method could affect the contents of tea polyphenols, catechin fractions, soluble sugars, free amino acids, caffeine, thearubigins, theabrownins and aroma substances of Congou black tea, which presented different concentrations, thus affecting the quality of Congou black tea, and there were different degrees of correlation among the internal components.” This part is just a resume about the parameters influence, its is not too a conclusive information about what was done.
- It is missing some conclusion about advantages and disadvantages of each methods to the tea characteristics (nutrition, color, antioxidant..)
- I also missed some quantitative data in conclusion, about the best used parameters to get the higher tea material preparation and flavors.
References
- Please, check all the references: There are too many references out of the journal rules!
-
- 6. Tea Research Institute, G.A.O.A.; Key Laboratory Of South China Agricultural Plant Molecular Analysis And Genetic Improve-433 ment Guangdong Provincial Key Laboratory Of Applied Botany, S.C.B.G.; Key Laboratory Of South China Agricultural Plant 434 Molecular Analysis And Genetic Improvement Guangdong Provincial Key Laboratory Of Applied Botany, S.C.B.G.; Center Of 435 Economic Botany, C.B.G.C.; Key Laboratory Of South China Agricultural Plant Molecular Analysis And Genetic Improvement 436 Guangdong Provincial Key Laboratory Of Applied Botany, S.C.B.G.; Tea Research Institute, G.A.O.A.; Tea Research Institute, 437 G.A.O.A.; Key Laboratory Of South China Agricultural Plant Molecular Analysis And Genetic Improvement Guangdong Pro-438 vincial Key Laboratory Of Applied Botany, S.C.B.G.; Center Of Economic Botany, C.B.G.C.; Tea Research Institute, G.A.O.A. 439 Enzymatic Reaction-Related Protein Degradation and Proteinaceous Amino Acid Metabolism during the Black Tea (Camellia 440 sinensis) Manufacturing Process. Foods. 2020, 9, 66.
8. Saptashish, D.; R., J.P.K. A Review of Withering in the Processing of Black Tea. Journal of Biosystems Engineering. 2016, 41, 365-444 372. 445
9. Key Laboratory Of Horticulture Plant Biology, M.O.E.C.; Key Laboratory Of Urban Agriculture In Central China, M.O.A.W.; 446 Key Laboratory Of Horticulture Plant Biology, M.O.E.C.; Key Laboratory Of Horticulture Plant Biology, M.O.E.C.; Key Labor-447 atory Of Horticulture Plant Biology, M.O.E.C.; Key Laboratory Of Horticulture Plant Biology, M.O.E.C.; Key Laboratory Of 448 Horticulture Plant Biology, M.O.E.C.; Key Laboratory Of Horticulture Plant Biology, M.O.E.C.; Key Laboratory Of Urban Agri-449 culture In Central China, M.O.A.W.; Key Laboratory Of Horticulture Plant Biology, M.O.E.C.; et al. Withering degree affects 450 flavor and biological activity of black tea: A non-targeted metabolomics approach. Lwt. 2020, 130.
12. Key Laboratory Of Horticultural Plant Biology, M.O.E.C.; Institute Of Fruit And Tea, H.A.O.A.; Institute Of Fruit And Tea, 456 H.A.O.A.; Key Laboratory Of Horticultural Plant Biology, M.O.E.C.; Key Laboratory Of Horticultural Plant Biology, M.O.E.C.; 457 State Key Laboratory Of Tea Plant Biology And Utilization, A.A.U.C.; Key Laboratory Of Horticultural Plant Biology, M.O.E.C.; 458 State Key Laboratory Of Tea Plant Biology And Utilization, A.A.U.C. Novel insight into the role of withering process in charac-459 teristic flavor formation of teas using transcriptome analysis and metabolite profiling. Food Chem. 2019, 272, 313-322.
18. Jinjin, W.; Wen, O.; Xizhe, Z.; Yongwen, J.; Yaya, Y.; Ming, C.; Haibo, Y.; Jinjie, H. Effect of shaking on the improvement of aroma 474 quality and transformation of volatile metabolites in black tea. Food Chemistry: X. 2023, 20, 101007.
19. Yang, S.; Pathak, S.; Tang, H.; De Zhang; Chen, Y.; Ntezimana, B.; Ni, D.; Yu, Z. Non-Targeted Metabolomics Reveals the Effects 476 of Different Rolling Methods on Black Tea Quality. Foods. 2024, 13.
21. G., N.P.P.K.; K., H.L.S.; A., D.P.S.J.; N., U.E.E.; S., J.W. Evaluation of inherent fructose, glucose and sucrose concentrations in tea 479 leaves (Camellia sinensis L.) and in black tea. Applied Food Research. 2022, 2.
Author Response
Dear Editor,
I would like to express my deepest gratitude to you for taking the time to review my manuscript and providing valuable feedback amid your busy schedule. Your suggestions have greatly benefited me, and I sincerely thank you for your guidance.
Yours sincerely, Yamin Wu
1 – Graphical abstract
Comments 1: I suggest to increase the figures size. they are too small.
Response 1: Dear respected editor, thank you for pointing this out. The revised graphical abstract has been reinserted into the current manuscript on page 2. Thank you again for your suggestion.
Material and methos
Comments 2:In order to be comparative, the leaf thickness size shouldnt be the same for all tested methods? I think the thickness will directly impact in area surface and contact and the dryness speed.
Response 2: Dear Editor! Thank you for your suggestion! It is true that different thickness of spreading will affect the area surface and contact and the dryness speed, since this study is based on 5 types of withering methods used in production, different withering methods correspond to their appropriate thickness of spreading. For example, the thickness of leaf spreading for warm-air withering is 10 cm because the warm air will make the water dissipate quickly, but if the thickness of leaf spreading is too thin, the water will be dissipated too quickly, the transformation of the inner substances will not be sufficient, and the edge of the fresh leaves may also become dry. natural withering in the whole withering process is more stable temperature and humidity, the fresh leaves slowly lose water, leaf spreading 3cm is more appropriate. Leaf spreading thickness is a variable in different withering methods, as are temperature, humidity and ventilation. Thanks again for your advice!
Comments 3 - what was the leaf thickness in the SHW? and the humidity of 65% was lasted during the whole process
Response 3:Dear editor, thank you very much for your reminder. The thickness of the spread leaf handled by SHW is 3cm. Since the SHW treatment is carried out indoors, the humidity is relatively stable but may vary slightly. In order to make the manuscript more rigorous, it has been changed to "around 65%". The changes are on lines 82, 93, and 94 on page 3 of the current manuscript. Thanks again for the reminder.
Comments 4 : Why you add 1g of sodium chloride and ethyl cinnamate in the headspace? Wht it means as bennefits to the sample extraction process?
Response 4:Dear editor, thank you for pointing out the question. This will make the methodology of our study clearer! Adding sodium chloride originally was to saturate the solution and prevent aromatic substances from dissolving in water. I have provided an explanation for this operation in the manuscript, and the additional content has been added on page 4, line 157 of the current draft. Thank you once again for your question!
Comments 5:Why have u chosen the DVB/CAR/PDMS fiber? Why not PDMS or DVB/CAR? Have u tested other fibers before to check how they work on your products?
Response 5:Dear editor, thank you for your question! The DVB/CAR/PDMS fiber represents three different stationary phase coatings with different polarities, which can adsorb aroma compounds of different natures and concentrations. The solid-phase microextraction needle used in this study is highly sensitive to alcohol, aldehyde, ketone, and alkane aroma compounds, which is in line with the types of aroma compounds found in tea. After multiple experiments, we found that this type of needle provides the best detection results.
Comments 6:Qualitative analysis was done using GC-MS or GC-FID?
Response 6:Dear editor, thank you for your question. In this study, qualitative analysis was performed using GC-MS.
Results:
Comments 1: Is there any information about the concentration of phenolic compounds before and after the withering process?
Response 1:Dear editor, thank you for your question, after this experiment we also found that taking stage samples can better illustrate the results, so based on this experiment, we have carried out further research, currently in the receipt analysis stage, hope to have the opportunity to get your guidance again. In this study, all the fresh leaves of the three replications were selected from the same tea plantation, and each time before processing was carried out to ensure that the quality of the fresh leaves of each treatment is relatively consistent, in this study, only the three replications of the 15 samples of the dry tea were tested for polyphenolic compounds.Please let us know if you require any further information. Thank you so much!
Comments 2: Why shaking process has the loewst and the SNW and WaW the highest polyphenolic content? what has impact the oxidation?
Response 2: Dear Editor, firstly, thank you for raising your concerns and I apologize for not expressing it clearly, causing confusion. In this study, SNW treatment showed the highest polyphenolic content, while WAW treatment showed the lowest, followed by SHW treatment. The reason for the lower content in SHW treatment is that during the withering process, the edges of the fresh leaves were damaged, which caused the polyphenolic substances in the vacuoles to come into contact with polyphenol oxidase, leading to premature consumption of tea polyphenols, resulting in a lower content. After conducting multiple literature searches, it was shown that warm-air withering resulted in lower polyphenol content and decreased bitterness in taste, however, the specific reason for this is still uncertain. Once again, thank you for raising this question, which may become the focus of our next project!
Comments 3:the information in this Figure is too pale light color, it is not good quality. I suggest to improve it.
Response 3:Dear editor, thank you for your valuable suggestions. We have darkened the colors for figures 1, 3, 5, and 7. Once again, thank you for your suggestions!
Comments 4 I really missed the chromatograms to compare the chemical profile. I think it is important to understand visually the impact of the different withering methods on chemical pro.
Response 4: Dear editor, thank you for your suggestion. We have attached the aroma chromatograms of all samples at the end of this reply. If you have any further requests, please let us know and we will gladly provide the relevant information. Thank you once again for your review!(GC-MS chromatograms are all attached in the PDF.)
Comments 5:Very interesting, I really thought that WAW could negatively impact the presence of monoterpene derivative compounds in the oils due to their higher volatility, how the essential oils are stored in the leaves? It seems they are well protected.
Response 5:Dear editor, thank you very much for your question. Monoterpene derivative compounds are very important for the aroma of black tea, such as linalool and geraniol. Initially, these substances exist in the form of glycosides, and during processing, they are hydrolyzed by glycosidase enzymes to appear in their free form.
Discussion
Comments 1: I think the discussion can be improved. I would expect some data about advantages and disadvantages of each method used taking different aspects in consideration like flavors (taste, aroma); colors; antioxidant properties, and loss of nutritional components by accelerate oxidation due to some withering process.
It is known that natural as well as some indoor withering methods tend to preserve higher levels of polyphenols, since in this process the leaves gradually lose moisture, while enzymes like polyphenol oxidase remain active. This fact is described to contribute to the tea’s astringency and antioxidant properties, which is interesting in some cases. On the other hand, outdoor and combined withering methods, could enhance the production of volatile aromatic compounds, giving the tea a stronger fragrance and richer flavor.
In some cases, like large amounts in industry, the mechanical hot-air withering is advantageous for mass production due to its efficiency, however it is reported to also compromise the nuanced qualities of the tea, since the high temperatures accelerate enzymatic reactions and moisture evaporation, it can reduce the complexity of the tea's aroma and lead to a loss of certain more volatile compounds, like monoterpene derivatives that can contribute to the subtlety of Congou black tea.
Due to this I would suggest to insert the HPLC chromatogram in the files to really see before and after how the withering process impact the chemical constituents in the tea composition. Also CG-MS of SPME to check about the aroma would be interesting.
Response 1: Dear editor, thank you very much for your careful and patient review and response to our manuscript. We have added relevant discussions on withered and natural withering in the corresponding section. The additional content can be found on pages 13 (lines 399-402) and 14 (lines 412-417) of the current manuscript. Currently, there is a lack of in-depth research on the effects of different withering methods on the quality of black tea. Most studies compare two or three withering methods, and the results are highly correlated with the climate of the tea-growing region and tea tree varieties. There may be some variations in the presentation of the results, but the majority are consistent. It is well known that withering is a key factor in the success of black tea production, so it is necessary to conduct further research based on this study. Our study explains the impact of different withering methods on the quality and content of tea components, providing a theoretical basis for selecting appropriate processing methods in Yunnan tea region. Some of the reasons behind these differences have been explored, but a comprehensive comparison of the five methods is still lacking. We have conducted preliminary tests and are currently in the data analysis stage. In light of your key suggestions and interesting viewpoints, we would be happy to conduct further analysis and research. We apologize again for not including the changes in polyphenolic substances before and after withering in this study. We genuinely appreciate your advice as it will be very helpful for the current research project we are working on.
Conclusion
Comments 1:Can be improved
Ex: “The results showed that the withering method could affect the contents of tea polyphenols, catechin fractions, soluble sugars, free amino acids, caffeine, thearubigins, theabrownins and aroma substances of Congou black tea, which presented different concentrations, thus affecting the quality of Congou black tea, and there were different degrees of correlation among the internal components.” This part is just a resume about the parameters influence, its is not too a conclusive information about what was done.
It is missing some conclusion about advantages and disadvantages of each methods to the tea characteristics (nutrition, color, antioxidant..)
I also missed some quantitative data in conclusion, about the best used parameters to get the higher tea material preparation and flavors.
Response 1: Dear editor, thank you for your suggestion! The conclusions of this manuscript have been revised accordingly, and the revised content can be found on pages 14, lines 434-442 of the current manuscript. Thank you once again for your feedback on the manuscript!
References
Comments 1:Please, check all the references: There are too many references out of the journal rules!
Response 1: Dear editor, I have made revisions to the citation format for the references [6], [8], [9], [12], [18], [19], and [21] in this manuscript. The modified content is highlighted in red. Thank you again for your patience in responding!
Dear Editor!
Thank you for carefully reviewing and patiently responding to my manuscript! I am also very grateful for the valuable suggestions you provided, as they are essential for improving the quality of this manuscript and have greatly inspired my future research. If there are any questions that I have not got you or any comments that I have overlooked, please let me know, and we will make the necessary improvements. Thank you once again! Best regards!
Yours sincerely, Yamin Wu

Reviewer 2 Report
Comments and Suggestions for Authors
-The authors have studied the impact of five different withering methods (NAW, WAW, SNW, SUW, SHW) on the quality of Congou black tea (sensory evaluation). The HS-SPME was used to determine the volatile compounds, and the chemical composition of aroma substances was analyzed using GC-MS indicating the presence of 227 components, concerning the amino acid the number of the constituents does not exceeds 22.
In order to discriminate between the effects of the five methods used, the authors have provided clustering heatmap analyses related to aroma components (227: alcohols (58), aldehydes (36), ketones (23), acid esters (59), hydrocarbons (42) and other substances (9)), amino acid fractions (22), as well as their impact on the composition of catechins (polyphenols) using Bubble diagram.
Comments
-In order to check the presence of the different chemical composition provided by the authors and to carry out a correct evaluation of the submitted-manuscript paper, the authors are invited to complete their submission by providing the missing GC/MS chromatograms (217 constituents) and HPLC chromatograms (polyphenols: catechins), as well as ion exchange chromatography related to the five different withering methods used (NAW, WAW, SNW, SUW, and SHW).
-Page 3 (2.2 Chemicals Part): many chemicals components are missing and must be provided such as ‘ethyl cinnamate in (page 4 , line 141), …
-Page 3 (line 113), ‘2.3. Instruments and Equipment’: ‘A lack of detail regarding the instrumentation used, such as HPLC (type of detector, column characteristics, mobile and stationary phases) as well as the data related to ion exchange chromatography
-Page 8 (lines 259-260), ‘whereas epigallocatechin gal-late and epigallocatechin gallate have both bitter and astringent tastes.’: Please avoid redundancy, the same word is repeated twice !
-What is the analytical technique used by the authors to determine the presence of the following antioxidants ‘catechins’ provided in the results part (3.4; Effects of … Catechins)? is it HPLC, did they use standards? If so, they needed to be clarified to make it easier for the reader to understand.
-The following techniques were introduced in the text (page 4, lines 117-120), however no corresponding experimental results were provided by the authors, we don't even talk about them in the text. It is not clear for the readers, and so they find themselves disoriented. Please, To make up for the shortfall.
. Page 4 (line 118), ‘1525 HPLC-type high-performance liquid chromatograph (American WA-TERS)’;
. Page 4(line 117), ’UV-2102PC UV‒VIS Spectrophotometer (Shanghai Element Analytical Instrument Co., Ltd.)’:
. Page 4(lines 119-120), ’QP2020NX gas chromatography‒mass spectrometry system (Shimadzu Corporation, Japan)’
-Page 4 (line 140): What do you means by the following writing ‘5 ul…’ ?
-Page 4 (line 155), ‘…gas elium,…’: To change with ‘…gas helium,…’
-Page 4 (line 146), ‘2.6.2. Analysis Conditions…’: Please add the following specific data to the GC-MS instruments used ‘QP2020NX gas chromatography‒mass 119 spectrometry system (Shimadzu Corporation, Japan).’
-The conclusion part needs to be further strengthened by relevant results found.
References Part:
- Please, the journal template must be respected, such as the following journal names must be abbreviated:
[8] …..’… Journal of Biosystems Engineering …’: Please, Journal name must be abbreviated.
[18] …..’… Food Chemistry …’: To change with ’… Food Chem.…’
[21] … ‘…Applied Food Research… ’ To rectify as following ‘…Appl. Food Res.… ’
-To rectify the writing of the authors name related to the five following references:
[6], [9], [12], [15], [34]
-Some grammatical English language errors must be corrected in the text. I suggest careful reading in order to avoid spelling errors or certain oversights.
-Page 4 (line 140) ‘at60°C and…’: Leave space ‘at60°C’.
-Page 8 (line 258), ‘…Sanderson et al.[24]…’: Leave space between ‘…al.[24]…’.
-Same remark for the line 260, ‘Scharbert et al.[25]’
Author Response
Dear Editor,
I would like to express my deepest gratitude to you for taking the time to review my manuscript and providing valuable feedback amid your busy schedule. Your suggestions have greatly benefited me, and I sincerely thank you for your guidance.
Yours sincerely, Yamin Wu
Common1:In order to check the presence of the different chemical composition provided by the authors and to carry out a correct evaluation of the submitted-manuscript paper, the authors are invited to complete their submission by providing the missing GC/MS chromatograms (217 constituents) and HPLC chromatograms (polyphenols: catechins), as well as ion exchange chromatography related to the five different withering methods used (NAW, WAW, SNW, SUW, and SHW).
Response1: Dear editor, first of all, I would like to express my respect for your rigorous research attitude. The chromatograms of the fragrance substances have been attached at the end of this email. However, I must apologize as the chromatograms of the catechin components were not retained because the data was obtained after peak calibration. Once again, I apologize for my negligence, but I can assure you that all the data is authentic. Thank you again for your questions, which will surely motivate me to cultivate better research habits.(GC-MS chromatograms are all attached in the PDF.)
Common2:Page 3 (2.2 Chemicals Part): many chemicals components are missing and must be provided such as ‘ethyl cinnamate in (page 4 , line 141),
Response2:Dear editor, thank you for carefully reviewing our manuscript! We have now supplemented the missing chemicals, and the added content can be found on page 3, lines 111-118 of the current manuscript. Thank you once again for your suggestions!
Common3:Page 3 (line 113), ‘2.3. Instruments and Equipment’: ‘A lack of detail regarding the instrumentation used, such as HPLC (type of detector, column characteristics, mobile and stationary phases) as well as the data related to ion exchange chromatography
Response3: Dear Editor, Thank you for your valuable suggestions! We have supplemented the issues you raised, and the added content can be found on pages 4, lines 125-129 of the revised manuscript. We greatly appreciate your feedback once again!
Common4: Page 8 (lines 259-260),whereas epigallocatechin gal-late and epigallocatechin gallate have both bitter and astringent tastes.’: Please avoid redundancy, the same word is repeated twice !
Response4: Dear Editor! Thank you for your question! I also apologize for the oversight. “epigallocatechin gallate and epigallocatechin gallate” was intended to mean “gallocatechin gallate and epigallocatechin gallate”. The modifications have been made in the manuscript, and the revised content can be found on page 9, line 288 of the current manuscript. Thank you once again for your careful review!
Common5:What is the analytical technique used by the authors to determine the presence of the following antioxidants ‘catechins’ provided in the results part (3.4; Effects of … Catechins)? is it HPLC, did they use standards? If so, they needed to be clarified to make it easier for the reader to understand.
Response 5: Dear editor! Thank you for your question! The detection of catechins was conducted using a standard sample through HPLC. Supplementary information regarding the standard sample, equipment, and methodology has been provided. The additional content is found on page 3, lines 116-118, and on page 4, lines 125-126 of the current manuscript.
Common6:The following techniques were introduced in the text (page 4, lines 117-120), however no corresponding experimental results were provided by the authors, we don't even talk about them in the text.It is not clear for the readers, and so they find themselves disoriented. Please, To make up for the shortfall.
Response 6:Dear editor! Thank you for your questions! The UV-2102PC UV‒VIS Spectrophotometer was used to detect caffeine, free amino acids, and soluble sugars, with the corresponding results provided in section 3.2. The 1525 HPLC-type high-performance liquid chromatograph was utilized for the detection of catechin components, with the corresponding results given in section 3.4. The Amino Acid Analyzer was employed for amino acid component analysis, and the results are presented in section 3.3. The QP2020NX gas chromatography‒mass spectrometry system, B13-3 Intelligent Constant temperature Timing Magnetic Stirrer, and 50/30 μm DVB/CAR/PDMS solid-phase microextraction fiber were used for aroma detection analysis, as explained in sections 2.6.1 and 2.6.2. The methodology section has already been supplemented. Once again, thank you for your guidance!
Common 7:Page 4 (line 140): What do you means by the following writing ‘5 ul…’ ?
Response 7:Dear editor, thank you for your questions! The detection of aroma is determined using the internal standard method, in which ethyl cinnamate is the internal standard, and the amount added is "5ul".
Common 8:Page 4 (line 155), ‘…gas elium,…’: To change with ‘…gas helium,…’
Response 8:Dear Editor, Thank you very much for raising the question and I apologize again for my oversight. The modifications have already been made in the manuscript, with the revised section found on page 5, line 172. Thank you again for your question!
Common 9:Page 4 (line 146), ‘2.6.2. Analysis Conditions…’: Please add the following specific data to the GC-MS instruments used ‘QP2020NX gas chromatography‒mass 119 spectrometry system (Shimadzu Corporation, Japan).’
Response 9: Dear editor,thank you very much for your suggestions! We have included the model number of the chromatographic column in section 2.3 and made the appropriate modifications in section 2.6.2. The revised content can be found on page 4, lines 128-129, and on page 5, lines 165-166 of the current manuscript. Thank you very much! Best regards.
Common 10:The conclusion part needs to be further strengthened by relevant results found.
Response 1: Dear editor, thank you for your suggestion! The conclusions of this manuscript have been revised accordingly, and the revised content can be found on pages 14, lines 434-442 of the current manuscript. Thank you once again for your feedback on the manuscript!
References Part:
Common 11:Please, the journal template must be respected, such as the following journal names must be abbreviated:
[8] …..’…Journal of Biosystems Engineering…’ Please, Journal name must be abbreviated.
[18] …..’…Food Chemistry …’:To change with ’…Food Chem.…’
[21] … ‘…Applied Food Research… ’To rectify as following ‘…Appl. Food Res.…
-To rectify the writing of the authors name related to the five following references:
[6], [9], [12], [15], [34]
Response 11:Dear editor,thank you very much for your feedback! We have made the abbreviations for the journal names of [8], [18], and [21] in the manuscript, as well as modified the authors’ names for the references [6], [9], [12], [15], and [34].The revised content has been highlighted in red in the reference section. We appreciate your careful review and once again, thank you for your suggestions.
Common 12:Some grammatical English language errors must be corrected in the text. I suggest careful reading in order to avoid spelling errors or certain oversights.
-Page 4 (line 140) ‘at60°C and…’: Leave space ‘at60°C’.
-Page 8 (line 258), ‘…Sanderson et al.[24]…’: Leave space between ‘…al.[24]…’.
-Same remark for the line 260, ‘Scharbert et al.[25]’
Response 12:Dear Editor, thank you for your careful review. I have added a space in 'at 60°C’. The revised content can be found on page 4, line 157 of the current manuscript. I have also added a space after [24] and [25]. The revised content can be found on page 9, lines 287 and 289 of the current manuscript. Thank you for your advice once again.
Dear Editor!
Thank you for taking the time to review my manuscript and providing valuable feedback! Your suggestions have greatly contributed to the improvement of this manuscript. If there are any questions that I have not got you or any comments that I have overlooked, please let me know, and we will make the necessary improvements. Thank you once again! Best regards!
Yours sincerely, Yamin Wu

Reviewer 3 Report
Comments and Suggestions for Authors
In this manuscript the authors investigate the effects of different withering methods on the quality of Gongou black tea. Overall the manuscript is understandable, however, on a broader point, I am concerned with the accuracy of the statistical methods used . I would greatly recommend the authors to give information about the statistical methods used, as well as its conditions of applicability. Regarding to the multiple comparisons test. With regard to the multiple comparisons test, this is not the most appropriate. We have multiple comparisons, so the Bonferroni test should be used.
The manuscrip needs revisions in the sections: Sensory evaluation, data analysis and results. I recommend that the manuscript be revised.
The authors should give more details about the Sensory evaluation. The information given in this section is not enlightening.
The section 2.7 needs to be improved. A more complete information is required. This section should mention the descriptive measures used, the graphical representations and methods used and their conditions of applicability. Regarding to statistical analysis the authors should give information about the assumptions of the methods used. They should provide more information when presenting statistical results. Saying that the test value is <0.05 is not enough. Therefore, the authors should state which technique they used and present the other values obtained, such as the observed value of the test statistic, the effect size, etc.
Line 22: Please correct “P<0.05” to “p<0.05”. When refering to p value we must write a small letter not a capital letter. Please correct that in all text.
Line 88: Please delete : ”.” after 50% and “in the morning”, because you have “11 a.m.”
Line 102: Please correct “20 min” to “20 minutes”
Line103: Please delete”.” after 25%
Section 2.4: Can the authors explain how the sensory evaluation is performed. The sensory characteristics are evaluated by numerical values? Did the experts used a likert scale from 1 to 5 or 1 to 10 to score the sensory characteristics? How many replicates were evaluated? Please explain the whole sensory evaluation process better.
Line137: Please delete ”.”
Line 140: please add space before “60ºC”
Line 166: Please correct: “single- factor analysis of variance” to “one way analysis of variance” also designate as one way ANOVA.
Line 172 to 183: Please correct “P<0.05” to “p<0.05”. In this paragraph the authors presented the following results “SHW (95.2±0.72),” without saying that this values represented the mean and the standard deviation. So I suggest that in the section 2.7 the authors included this information.
In section 3.1 authors said : “The total scores of SHW and NAW differed significantly from those of the other four withering methods (P<0.05)” But they only give the p value. Where are the other statistical results? What was the statistical analysis. Before presenting the results the authors should informed what statistical analysis was performed and then give the results. They are talking about aroma and taste, So It was a MANOVA? Were are the results of the MANOVA? And the ANOVA results? They only give the p value. The authors should mention the technique used and present not only the test values, but also the other results, such as the observed values of the test statistic and the effect size. And in the case of multiple comparison tests, in addition to the test values, they should also present the confidence intervals.
Lines 196, 199, 205 and 209: Please correct “P<0.05” to “p<0.05”. Where are the others statístical results. As I said earlier, the presentation of the p value is not enough.
Line 213: Please correct “P<0.05” to “p<0.05”.
Section 3.3 and following: The authors should mention the technique used and present not only the test values, but also the other results, such as the observed values of the test statistic and the effect size.
As mention before the the recommendations made with regard to statistical analysis apply throughout the document.
Author Response
Dear Editor,
I would like to express my deepest gratitude to you for taking the time to review my manuscript and providing valuable feedback amid your busy schedule. Your suggestions have greatly benefited me, and I sincerely thank you for your guidance.
Yours sincerely, Yamin Wu
Responses
Comments 1:In this manuscript the authors investigate the effects of different withering methods on the quality of Gongou black tea.Overall the manuscript is understandable, however, on a broader point, I am concerned with the accuracy of the statistical methods used .I would greatly recommend the authors to give information about the statistical methods used, as well as its conditions of applicability. Regarding to the multiple comparisons test. With regard to the multiple comparisons test, this is not the most appropriate. We have multiple comparisons, so the Bonferroni test should be used.
Response 1: Dear Editor, thank you for your valuable suggestions and new perspectives. In this study, analysis of variance and the Least Significant Difference test were used to evaluate the significance of the differences in compound content of black tea under different withering methods. Analysis of variance is commonly used to compare the effects of different processing methods on tea quality, and it is frequently employed in tea studies. Perhaps there was an issue with translation accuracy, and we have already made revisions to address this. We appreciate your suggestion of using the Bonferroni test. However, as tea leaves undergo processing, the variables are not independent but rather have inherent relationships. Moreover, as we have limited knowledge about this particular method, we have extensively searched relevant literature related to our manuscript but have not found any studies that use this method. We apologize for our lack of knowledge in this area. We are currently intensifying our learning efforts and are grateful for the new method you have provided!
Comments 2:The manuscript needs revisions in the sections: Sensory evaluation, data analysis and results.I recommend that the manuscript be revised.
The authors should give more details about the Sensory evaluation. The information given in this section is not enlightening.
Response 2:Dear editor, thank you for your valuable suggestions! In order to present the results and details of the sensory evaluation more clearly and help readers better understand, we have made additions to sections 2.4 on page 4. The additional content can be found between lines 136-141. Furthermore, we have added a sensory evaluation table on page 6 and provided explanations for the evaluation table between lines 212-215. We would appreciate it if you could review these changes, and if there are any further questions or issues, we would be happy to provide additional information. Thank you again!
Comments 3:The section 2.7 needs to be improved.A more complete information is required.This section should mention the descriptive measures used, the graphical representations and methods used and their conditions of applicability. Regarding to statistical analysis the authors should give information about the assumptions of the methods used. They should provide more information when presenting statistical results. Saying that the test value is <0.05 is not enough. Therefore, the authors should state which technique they used and present the other values obtained, such as the observed value of the test statistic, the effect size, etc.
Response 3: Dear editor, first of all, thank you for raising the question and I apologize for my inaccurate statement. I have made changes to section 2.7 and included the TBtools plotting tool, which presents the analysis results of aroma substances very well. The reason for conducting variance analysis is that both the detection of internal components and sensory evaluation were performed in 3 parallel experiments. The least significant difference test was used to analyze the significance of compound content in different withering methods. The revised content can be found on page 5, lines 182-185. Thank you once again!
Comments 4:Line 22:Please correct“P<0.05” to “p<0.05”. When refering to p value we must write a small letter not a capital letter. Please correct that in all text.
Response 4: Dear editor, thank you for your reminder! I apologize for my negligence and have made the necessary corrections to the entire manuscript regarding the “P” issue. Thank you once again for pointing it out!
Comments 5:Line 88: Please delete : ”.” after 50% and “in the morning”, because you have “11 a.m.”
Response 5:Dear editor, thank you for your careful review! The necessary revisions have been made to this question, and the updated content can be found on page 3, line 87. Thank you once again!
Comments :6:Line 102: Please correct “20 min” to “20 minutes”.
Response 6:Dear editor, thank you for your correction! I have revised “20 min” to "20 minutes". The modified content can be found on page 3, line 102. Thank you once again!
Comments :7:Line103: Please delete”.” after 25%
Response 7: Dear editor, thank you for your correction! I have removed the “.” and the revised content can be found on page 3, line 103 of the current manuscript. Thank you once again!
Comments 8:Section 2.4: Can the authors explain how the sensory evaluation is performed. The sensory characteristics are evaluated by numerical values? Did the experts used a likert scale from 1 to 5 or 1 to 10 to score the sensory characteristics? How many replicates were evaluated? Please explain the whole sensory evaluation process better.
Response 8:Dear editor, thank you for your question! The sensory evaluation in this experiment is conducted according to GB/T 23776-2018 “Methodology for sensory evaluation of tea” in China. The sensory evaluation mainly focuses on the appearance of dry tea, liquor color, aroma, taste, and infused leaf of black tea, with each factor scored out of 100 points. Each factor has a different weight in the total score (out of 100), and the weights are supplemented in Table 1. The specific procedure of this experiment involves 7 expert evaluators divided into 3 groups, evaluating the appearance of dry tea first, then weighing 3g of sample in a white porcelain cup (150ml), adding 100°C hot water, brewing for 5 minutes, pouring out the tea liquor in order to evaluate and score the liquor color, aroma, taste, and infused leaf of black tea. The higher the average total score, the better the quality. The corresponding operation process has been supplemented in 2.4. Thank you again for your thorough review of my manuscript! If there are any other areas that need to be modified, please do not hesitate to let me know. Thank you once again for your question!
Comments 9:Line137: Please delete ”.”
Response 9: Dear editor, thank you for your correction! The “.” has been removed, and the revised content can be found on page 4, line 153 of the current manuscript. Thank you again for your guidance.
Comments 10:Line 140: please add space before “60ºC”
Response 10:Dear Editor, I would like to express my sincere gratitude for your meticulous review of my manuscript. I have now added spaces in between "60℃". Thank you once again for your assistance!
Comments 11:Line 166: Please correct: “single- factor analysis of variance” to “one way analysis of variance” also designate as one way ANOVA.
Response 11:Dear Editor, thank you for your correction. Due to the inaccurate expression in our original manuscript, we have made revisions and retranslations accordingly. The modified content can be found on pages 5, lines 183-185 of the revised manuscript. Once again, we appreciate your valuable input.
Comments 12:Line 172 to 183:Please correctP<0.05” to “p<0.05”. In this paragraph the authors presented the following results “SHW (95.2±0.72),” without saying that this values represented the mean and the standard deviation. So I suggest that in the section 2.7 the authors included this information.
Response 12: Dear editor, thank you for your suggestion. We have made the necessary changes to all instances of “P” throughout the entire manuscript. To enhance the clarity of the manuscript, we have provided appropriate explanations for "±". The additional content can be found on page 5, line 182.
Comments 13:In section 3.1 authors said : “The total scores of SHW and NAW differed significantly from those of the other four withering methods (P<0.05)” But they only give the p value. Where are the other statistical results? What was the statistical analysis. Before presenting the results the authors should informed what statistical analysis was performed and then give the results. They are talking about aroma and taste, So It was a MANOVA? Were are the results of the MANOVA? And the ANOVA results? They only give the p value. The authors should mention the technique used and present not only the test values, but also the other results, such as the observed values of the test statistic and the effect size. And in the case of multiple comparison tests, in addition to the test values, they should also present the confidence intervals.
Response 13: Dear editor, thank you for your question! We also realized that incomplete information provided by ourselves has led to readers’ misunderstanding of 3.1. To address this issue, we have created Table 1 to explain the origins of various scores and the total score. The p-values are included to demonstrate whether different methods of withering significantly affect the total score. Relevant information can be found on page 4, lines 136-141 of section 2.4; page 5, lines 182-185 of section 2.7; and on page 6 in Table 1 and its accompanying notes. Once again, we appreciate your valuable feedback!
Comments 14:Lines 196, 199, 205 and 209: Please correct “P<0.05” to “p<0.05”. Where are the others statístical results. As I said earlier, the presentation of the p value is not enough.
Response 14:Dear Editor, thank you for your suggestions. In this study, we conducted tests on various components that influence the quality of black tea in order to analyze the effects of different withering methods on the quality and quantity of these components. By analyzing the p-values, we were able to understand the differences in component levels between different withering methods. Combining our study with relevant literature, we explored and discussed the reasons behind these effects. We also appreciate your suggestion, as it will assist me in learning new statistical analysis techniques. Thank you again for your guidance. Best regards!
Comments 15: Line 213:Please correct P<0.05” to “p<0.05”
Response 15:Dear Editor, thank you for your careful review. I have made the necessary revisions to “P.” The modified content can be found on page 7, line 242 of the current manuscript. I wish you all the best!
Comments 16: Section 3.3 and following: The authors should mention the technique used and present not only the test values, but also the other results, such as the observed values of the test statistic and the effect size.
Response 16: Dear editor, thank you for your suggestions!In Section 3.3, the content and differences in amino acid composition in different black tea withering methods are presented. The technique used for analyzing the components and their content is the amino acid autoanalyzer. The analysis reveals the content of amino acids that contribute significantly to the flavor of black tea and the differences in their content among different withering methods. This provides an understanding of the impact of different withering methods on the various amino acids. Best regards!
Comments 17: As mention before the the recommendations made with regard to statistical analysis apply throughout the document.
Response 17: Dear Editor, thank you for taking the time to review my manuscript and providing valuable feedback! Your suggestions have greatly contributed to the improvement of this manuscript. If there are any questions that I have not got you or any comments that I have overlooked, please let me know, and we will make the necessary improvements. I would also like to express my sincere apologies for my shortcomings in statistical analysis. Moving forward, we will continue to learn and explore new methods of comparative analysis. Thank you once again!
Dear Editor,
I would like to express my sincere gratitude for your patient review and valuable suggestions on my manuscript. I appreciate your time and effort. Wishing you all the best.
Best regards, Yamin Wu

Round 2
Reviewer 2 Report
Comments and Suggestions for Authors
Comments (revised manuscript-text )
1- In the references part: all the erroneous references provided by the authors were corrected according to the template of the journal.
2- Concerning the GC chromatograms presented separately: The reviewer finds a lot of difficulty and problems in distinguishing and discriminating between the different chromatograms provided because there is a lack of titles specific to each chromatogram or probably are given in Chinese language which is the reviewer is not supposed to know. The reviewer regrets not being able to decipher them in order to properly evaluate them.
3- Page 4 (line 185): is it ‘5 ul…’ or ‘5 microliter ‘(5 μl)’ ?
4- Page 3 (lines 116-117): Please in the revised version avoid the redundancy of the following antioxidant name which is ‘….epigallocatechin gallate, and epigal-locatechin gallate …’ repeated twice time !
5- Page 4 (lines 124-126), ‘2.3. Instruments and Equipment’: ‘A lack of detail concerning the HPLC mobile phase used, as well as the data related to ion exchange chromatography.
Author Response
Dear Editor,
Thank you very much for taking the time to provide feedback on this manuscript again. I apologize for not fully addressing the issues in the first round of revisions. I appreciate your patience in reviewing it. Wishing you all the best!
Yours sincerely, Yamin Wu
Common1: In the references part: all the erroneous references provided by the authors were corrected according to the template of the journal.
Response1:Dear Editor, thank you for your patience and careful guidance in reviewing the references.
Common2: Concerning the GC chromatograms presented separately: The reviewer finds a lot of difficulty and problems in distinguishing and discriminating between the different chromatograms provided because there is a lack of titles specific to each chromatogram or probably are given in Chinese language which is the reviewer is not supposed to know. The reviewer regrets not being able to decipher them in order to properly evaluate them.
Response2:Dear editor, I apologize for not providing explanations for each gas chromatography graph. I have now added explanations below each gas chromatography graph. All detected substances have been appended to the end of the reply. Please check and confirm receipt. Once again, thank you for your review.
Common3:3-Page 4 (line 185): is it ‘5ul…’ or ‘5microliter‘(5μl)’ ?
Response3: Dear editor, thank you very much for your questions, and I apologize for not making the correct changes the first time. I have now made the correction, and the revised content can be found on page 4, line 163 of the current manuscript. Thank you again for your careful review.
Common4: Page 3 (lines 116-117): Please in the revised version avoid the redundancy of the following antioxidant name which is ‘….epigallocatechin gallate, and epigal-locatechin gallate …’ repeated twice time !
Response4: Dear editor, first of all, thank you for carefully reviewing and raising the issue with this manuscript. These are indeed two different substances, and I apologize for the error in my writing that caused you to spend extra time on this. I have now made the necessary changes. The revised content can be found on page 3, line 116 of the current manuscript.Wishing you all the best!
Common5:5- Page 4 (lines 124-126), ‘2.3. Instruments and Equipment’: ‘A lack of detail concerning the HPLCmobile phaseused, as well as the data related to ion exchange chromatography.
Response 5: Dear Editor, thank you for your questions and I apologize for not providing a complete supplement for the first time. I have now supplemented the model of the HPLC, the chromatographic column model, the UV detector, mobile phase A, mobile phase B, column temperature, and flow rate. The supplemental content can be found on pages 4, lines 126-127 and 151-155 of the current manuscript. Wishing you all the best!
Dear Editor!
Thank you again for taking the time to carefully review this manuscript and provide valuable suggestions for improving this paper. Best regards!
Yours sincerely, Yamin Wu

Reviewer 3 Report
Comments and Suggestions for Authors
Dear authors
Regarding the article entitled Effects of Five Different Withering Methods on the Composition and Quality of Congou Black Tea, I still have some concerns regarding the experimental design and the statistical analysis. Therefore, I kindly request that you address the following questions:
In section 2.4 the authors stated:
“The tea tasting panel was composed of 7 experienced experts and were divided into 3 groups”. Can you explain that? Why did you divide the 7 experts into 3 groups? What was the purpose? Is the panel made up of men and women? how old are they? Author should characterise the tasting panel.
Regarding the evaluation of sensory characteristics: What methodology was employed? Was it a Quantitative Descriptive Analysis (QDA)? Additionally, what type of scale was used for scoring? Was it a Likert scale? If so, what were the score ranges—1 to 5, 1 to 10, or another scale? This information is essential for a comprehensive understanding and interpretation of the results. Please provide clarification on these points.
Upon reviewing the article, it is evident that various statistical methodologies have been applied; however, the authors fail to mention them in the Statistical Analysis section. Therefore, I recommend that the authors revise Section 2.7 to include all statistical methodologies used, provide a rationale for their application, and include details regarding the conditions for their applicability. This will enhance the clarity and rigor of the analysis presented.
In section Results the authors stated:
“Different withering methods have a greater impact on the aroma and taste of Congou black tea, and a smaller impact on its appearance, liquor color and infused leaves; however, the liquor color of WAW black tea is yellowish. The total scores of SHW and NAW differed significantly from those of the other four withering methods (p<0.05), and the highest total sensory scores were obtained for SHW (95.2±0.72), followed by NAW (93.7±0.34)…..”
What statistical technique was employed in the analysis? Where are the detailed statistical results? Simply stating "p < 0.05" is insufficient for proper interpretation. The authors should specify which statistical technique was used to address the research question and present the relevant statistical results, such as the observed value of the test statistic, p value and other key parameters. For instance, when using ANOVA, it is common practice to report the observed value of the test statistic along with its degrees of freedom, the p-value, and occasionally the effect size.
Therefore, I recommend that the authors include this information for all statistical results reported throughout the article (e.g., Sections 3.1, 3.2, 3.3, etc.). This will ensure the results are clearly presented and interpretable by the readers.
Table 1: Can the authors explain table 1? Explain the header and the results.
Upon reviewing the table, it is evident that some sensorial characteristics display zero variability values. Were these characteristics excluded from the statistical analysis? If not, how were they treated in the analysis? Please provide clarification.
Sections 3.5 and 3.6
Where is the information regarding the statistical techniques employed, and where are the detailed statistical results? The authors only report "p < 0.05," which is insufficient. The statistical results are not presented adequately. The authors need to specify the statistical techniques used and provide a complete set of results, including the observed value of the test statistic, degrees of freedom, and effect sizes where applicable. This level of detail is important for the proper evaluation and understanding of the analysis.
Author Response
Dear Editor,
Thank you very much for taking the time to provide feedback on this manuscript again. I apologize for not fully addressing the issues in the first round of revisions. I appreciate your patience in reviewing it. Wishing you all the best!
Yours sincerely, Yamin Wu
Responses
Comments 1:In section 2.4 the authors stated:“The tea tasting panel was composed of 7 experienced experts and were divided into 3 groups”. Can you explain that? Why did you divide the 7 experts into 3 groups? What was the purpose? Is the panel made up of men and women? how old are they? Author should characterise the tasting panel.
Response 1: Dear Editor, thank you very much for your questions. The tasting panel consists of 4 female experts and 3 male experts, aged between 30 and 50. The reason for dividing the tasting panel into three groups is as follows: the expert groups of two people each, where one person is the main scorer and the other expert primarily corrects the scores and raises any objections to the scoring. The three-member expert group represents a collective scoring method, as although the panel experts have professional certification, they still may have some subjectivity. Therefore, adopting multiple forms of review with different groups is to reduce subjectivity. Additionally, it allows calculation of the standard deviation of the three groups’ observations. If the standard deviation is large, it indicates a difference of opinions among the experts in scoring this factor. In the sensory evaluation of this experiment, the three expert groups had relatively consistent opinions on the scoring of each factor.Once again, thank you for your suggestions. The relevant information has been supplemented in Section 2.4! The revised content can be found on page 4, line 138 of the current manuscript. Thank you for your guidance!Best regards!
Comments 2:Regarding the evaluation of sensory characteristics: What methodology was employed? Was it a Quantitative Descriptive Analysis (QDA)? Additionally, what type of scale was used for scoring? Was it a Likert scale? If so, what were the score ranges—1 to 5, 1 to 10, or another scale? This information is essential for a comprehensive understanding and interpretation of the results. Please provide clarification on these points.
Response 2: Dear Editor, thank you very much for your question! In the sensory evaluation of black tea, GB/T 23776-2018 “Methodology for sensory evaluation of tea” of China has a clear score interval. How to determine the score of a factor within the score interval is also specified, such as the performance of a factor is slightly higher than the standard, then add 1 point, when the performance is higher, then add 2 points, significantly higher then add 3 points. According to the specific scoring requirements of “Methodology for sensory evaluation of tea” of China, the reviewing experts will combine their own specific situation of seeing, smelling, tasting and touching and then score. All scores in this sensory review are based on the scoring requirements of Methodology for sensory evaluation of tea of China. In this manuscript, the most important and basic procedures of the sensory evaluation and the scores of each factor are explained, but since each factor has its own scoring range and specific scoring rules, it would take quite a long and systematic text to explain each item. Dear editor, thank you in particular for your careful review of this paper, and we hesitate to add a systematic description of the scoring rules as they have not been presented in the literature we have seen so far. Once again, thank you for your valuable suggestions!
Comments 3:Upon reviewing the article, it is evident that various statistical methodologies have been applied; however, the authors fail to mention them in the Statistical Analysis section. Therefore, I recommend that the authors revise Section 2.7 to include all statistical methodologies used, provide a rationale for their application, and include details regarding the conditions for their applicability. This will enhance the clarity and rigor of the analysis presented.
Response 3: Dear Editor, Thank you for your question. In this manuscript, all the experiments were conducted three times, so the values are presented as the mean, and the standard deviation analysis was performed at the initial data processing stage to ensure that they were within the specified deviation range, or else their contents would need to be re-determined. Subsequently, the one-way ANOVA test and the least significant difference test were conducted for some internal components such as tea polyphenols, caffeine, free amino acids, etc., in order to identify the effects of different withering methods on the content of certain internal components. These are the analytical methods used in addition to the data processing (e.g., normalization) that comes with the plotting software. Thank you again for your question!
In section Results the authors stated:
Comments 4:“Different withering methods have a greater impact on the aroma and taste of Congou black tea, and a smaller impact on its appearance, liquor color and infused leaves; however, the liquor color of WAW black tea is yellowish. The total scores of SHW and NAW differed significantly from those of the other four withering methods (p<0.05), and the highest total sensory scores were obtained for SHW (95.2±0.72), followed by NAW (93.7±0.34)…..”
What statistical technique was employed in the analysis? Where are the detailed statistical results? Simply stating "p < 0.05" is insufficient for proper interpretation. The authors should specify which statistical technique was used to address the research question and present the relevant statistical results, such as the observed value of the test statistic, p value and other key parameters.For instance, when using ANOVA, it is common practice to report the observed value of the test statistic along with its degrees of freedom, the p-value, and occasionally the effect size.
Therefore, I recommend that the authors include this information for all statistical results reported throughout the article (e.g., Sections 3.1, 3.2, 3.3, etc.). This will ensure the results are clearly presented and interpretable by the readers.
Response 4: Dear Editor, thank you for your question. In section 3.1 of the results, we first performed standard deviation analysis on the rating scores of the 3 groups of evaluators. Then, we conducted one-way ANOVA and post hoc tests for significant differences. We have now made revisions to Table 1, where the specific scores represent the mean values of each factor’s observed values, and the data after ± represents their standard deviations. Different letters for the same factor indicate a significant difference between groups (p < 0.05). In this study, there are 4 degrees of freedom for between-group variability and 10 degrees of freedom for within-group variability. In the results section, the observed values are presented in the form of graphs, and corresponding textual explanations are provided for any parts not shown in the figures. Taking into account your suggestions, we have made the necessary modifications to Table 1 (page 6) and section 3.1. The revised content can be found on pages 5 (lines 198-200) and 6 (lines 223-224) of the current manuscript. Thank you again for your careful review of this paper.
Comments 5:Table 1: Can the authors explain table 1? Explain the header and the results.
Upon reviewing the table, it is evident that some sensorial characteristics display zero variability values. Were these characteristics excluded from the statistical analysis? If not, how were they treated in the analysis? Please provide clarification.
Response 5: Dear Editor, thank you for your question. Regarding the statement "it is evident that some sensorial characteristics display zero variability values", it is indeed referring to the standard deviation of the color scores for tea soup in the SHW treatment. The three groups of evaluators assigned the same score to the tea soup color, resulting in a standard deviation of 0. These data were included in the analysis, and a one-way analysis of variance and least significant difference test were performed on the 5 groups of data. The degrees of freedom for between-group variation were 4, and for within-group variation were 10. Thank you once again for your question.
Comments :6:Sections 3.5 and 3.6
Where is the information regarding the statistical techniques employed, and where are the detailed statistical results? The authors only report "p < 0.05," which is insufficient. The statistical results are not presented adequately. The authors need to specify the statistical techniques used and provide a complete set of results, including the observed value of the test statistic, degrees of freedom, and effect sizes where applicable. This level of detail is important for the proper evaluation and understanding of the analysis.
Response 6: Dear Editor, Thank you for your question. In Section 3.5, the average value of the observed data is presented through a line graph, with the deviation of the data being represented by error bars. A variance analysis and least significant difference test were performed on the components in Section 3.5, and a textual explanation was given for the substances that showed significant differences (p<0.05). Additionally, the reasons for these differences and possible influencing factors were explained, providing insight into the effect of different withering methods on the aforementioned substances. In section 3.6, we mainly present the proportions of different aroma compounds in black tea under different withering methods through stacked bar charts (i.e. observed values). The effects of withering methods on the categories of aroma compounds were also determined through analysis of variance and least significant difference tests. Since the impact of aroma compounds on the aroma profile is not only related to their content, but also closely related to their ROAV values, simply put, the size of the ROAV values determines whether these aroma compounds can be perceived. The calculation of ROAV values involves a series of conversions, and their magnitudes and relationships are presented in the form of a clustering heatmap. In all the above analyses, there are a total of 5 withering methods and 5 variables, and the presentation of each withering method’s components is based on the average of three experiments. The between-group variation has 4 degrees of freedom, while the within-group variation has 10 degrees of freedom. The observed values are mainly presented through related figures. Dear editor, thank you for your suggestions and for your patience in reviewing this manuscript!
Dear Editor,
I would like to express my sincere gratitude for your patient review and valuable suggestions on my manuscript. I appreciate your time and effort. Wishing you all the best.
Best regards, Yamin Wu
